# The Effect of Internal Pressure on Radial Strain of Steel Pipe Subjected to Monotonic and Cyclic Loading

**DOI:** 10.3390/ma12182849

**Published:** 2019-09-04

**Authors:** Costel Pleșcan, Mariana D. Stanciu, Matyas Szasz

**Affiliations:** 1Department of Civil Engineering, Transylvania University of Brasov, 500036 Brașov, Romania; 2Department of Mechanical Engineering, Faculty of Mechanical Engineering, Transylvania University of Brașov, B-dul Eroilor 29, 500036 Brașov, Romania

**Keywords:** steel pipe, monotonic load, cyclic load, internal pressure, radial strain

## Abstract

Steel pipes in different engineering applications may fail, leading to numerous environmental disasters. During loading, certain mechanical and chemical phenomena develop inside the pipes and cause them to burst. In this study, the influence of internal pressure on the elastic and plastic behaviour of E355 steel pipes was investigated on small specimens with different wall thicknesses. First, the failure modes of pipes subjected to monotonic loading were assessed, and then the behaviour of specimens subjected to cyclic internal pressure was analysed in terms of variation of radial strain. The strain and stress states of pipes were analysed using the finite element method. The results revealed that the hardening of materials inside the pipes increases the risk of cracking and bursting because of elasticity limits being exceeded, causing entry into the plastic domain. The transition of mechanical behaviour can be observed in the microstructure of steel in cracked areas from the inside to the outside of the pipe.

## 1. Introduction

In energy infrastructures, the critical assessment of the bursting pressure of pipelines is very important with regard to their cost-effectiveness, safety and integrity. So, the engineering challenge is to establish the proper dimensions of the external diameter and wall thickness. These dimensions depend on the rate of supply, internal pressure and the mechanical properties of the pipe’s materials. 

Depending on the location of the pipes (above the soil, in the ground or under water), there are several theories regarding the analysis of their elastic–plastic behaviour. Thus, for pipelines installed above the ground, some engineers use Timoshenko and Goodier’s elastic behaviour theory for the purpose of sizing pipes and choosing the material. Other design engineers rely on the theory of elastic–plastic behaviour. Prantdl and Reuss believe that total strain consists of an elastic and a plastic component [1]. Numerous studies on the elastic–plastic behaviour of pipes subjected to internal pressure led to the development of theories for predicting the burst failure at plastic collapse for pipes [2,3,4,5,6]. Zhu and Leis introduced into a study the plastic flow effect concerning the elastic–plastic response of pipes, which is based on the maximum shear stresses [3]. The geometrical parameters of critical axial cracking of pipelines subjected to internal pressure were assessed [7] using three analytical methods: the failure assessment diagram, Gauss–Seidel method and Folin–Ciocalteu method. Because the pipe failure occurs from the inside to the outside and the crack propagates along the pipe, the periodic verification of the pipe’s deformations constitutes a means by which to avoid the occurrence of a burst [2,5,8]. Some studies investigated the overlapping effects of cyclic stresses and corrosion failure on the lifetime of pipelines [9,10,11,12]. So, the root cause of the pipe failure consists of erosive wear, which leads to a reduction in the thickness of the pipe to a point at which failure occurs, as shown by [9]. From a mechanical point of view, it is important to take into account all causes that affect the appropriate functionality of pipes. The analytical and numerical methods can predict the damage models of pipes, but the experiments can offer more realistic failure behaviour [13,14]. The rheological behaviour of small specimens of steel pipe and the material’s microstructure in the failure area have been analysed based on experiments [13]. Other studies examine the strain evolution of pipes subjected to periodical cycles of internal pressure, in order to anticipate their behaviour over time [14]. In the literature, there are numerous theories in predicting the behaviour of pipes depending on the thickness of the walls (thin-wall pipes/thick-wall pipes), such as the von Mises yield criterion and Hill criteria, Tresca yield criterion, and depending on the rheology of materials from which they are made, the elastic–plastic criterion and plastic criterion [1,2,3,15,16,17].

The experiments on cyclic loads of pipe specimens are scarce in comparison to studies on monotonic behaviour. The cyclic plastic behaviour and the estimation of yield limits and hardening behaviour of metals from pipe structures is very complex, and experimental studies are required in order to cover several aspects. In this paper, the damage models and strains and stresses of steel pipes with different wall thicknesses were assessed after being subjected to high internal pressure in static and cyclic modes. The deformation in the radial direction was determined by measuring the external diameters of samples, before and after the internal pressure was applied. 

## 2. Materials and Methods

### 2.1. Sample Descriptions

The tested specimens were obtained from honed pipes with dimensions as presented in Table 1. The samples differed in wall thickness, which varied between 1.105 and 2.505 mm. The inner diameter was the same for all samples. The material of the samples was E355 steel with the following chemical composition: C: 0.140%; Si: 0.340%; Mn: 1.330%; P: 0.007%; S: 0.019%; Al: 0.026%. The mechanical properties of E355 steel are admissible yield stress (σ_c_) of 548 MPa and maximum stress (σ_r_) of 722 MPa. Table 1 presents the geometrical features of the samples. The shape and design of the specimens was executed using a computer numerical control machine according to EN 10305–1:2010 (Figure 1). The specimens received two types of tests: one group was subjected to a maximum internal pressure of 800 bar (80 MPa), and the other group was subjected to 10 loading cycles of 400 bar (40 MPa) internal pressure for 60 s alternating with an unloading period.

### 2.2. Experimental Setup

#### 2.2.1. Static Loading

In the first stage of the experiment, each small pipe specimen (1) was subjected to the same internal pressure, as provided by a hydraulic oil regime, and pressure from 100 to 1000 bars was measured on the exterior diameter with an electronic calliper (9) in order to observe the elastic–plastic deformation. The experimental setup consisted of an electric pump (8) with the electric current consumption of 4.5 A, working at a 50 Hz frequency and a maximum pressure of 800 bars, which provided the hydraulic oil regime. The pump transmitted the oil under pressure through a hydraulic hose (4) made in conformity with European standards (730 EN 856), having four metallic insertions (4SP) with a nominal diameter of 6 mm (DN6) resisting up to 450 bars and with length (*l*) = 2000 mm. To control the experiment, a faucet (6) (type KHB–D/4’’) with a nominal diameter of 6 mm (DN6) and a nominal pressure of 500 bars (PN500) was connected. A supplementary manometer of 1000 bars (7) was attached using a T-shaped connector of 1/4’’ (5). The connection between the manometer and samples was assured by the same type of hydraulic hose (4). At the end of each sample, a metallic cork with gasket was attached (3) (Figure 2). The experimental setup was based on a previous study [15], which considered that due to the experimental setup and specimen configuration, i.e., closed-end pipe, the internal pressurisation of the pipe resulted in the development of mainly hoop strain with a relatively smaller axial strain (due to Poisson’s effect). This was the reason for taking into account just the radial behaviour of the tested mini pipes. 

While testing, the specimens were contained inside a protective pipe to avoid accidents. The damage structure and the microstructure of steel in cracked areas were analysed using a digital microscope. 

#### 2.2.2. Cyclic Loading

Based on the ultimate strength of the pipe specimens as determined in the previous section, three types of specimens were subjected to zero-tension cycles of internal pressure of 400 bars (40 N/mm^2^). Using similar equipment for monotone loading of samples, in this case, the experimental stand was improved with a pressure-regulating valve, and the pump was set to 400 bars, which was operated by remote control for automatically increasing pressure in the pipe and depressurizing the sample for 30 s/cycle (for the first ten cycles) and 60 s/cycle (in the second stage) over ten cycles (Figure 3). In this test, three types of samples were investigated in accordance with the ratio between the outer and inner diameters, denoted β: sample A, with β = 1.120; sample B, with β = 1.104; and sample C, with β = 1.088. The geometrical features of the samples used in cyclic testing are presented in Table 2. The experimental setup for cyclic loading can be observed in Figure 4.

## 3. Results and Discussion

### 3.1. Monotonic Load

During the application of monotonic load, gradually increasing the internal pressure, the exterior diameter was measured for each specimen and the values were recorded, as presented in Table 3. The effect of internal pressure on radial strain and stress differs in accordance with the ratio β. In comparison to specimens with a ratio between the inner and outer diameter of more than 1.1 (as for samples 1 to 7), which resisted high pressures of up to 80 MPa, sample 8, with the ratio β = 1.1, failed at 75 MPa, and samples 9 and 10, with β < 1.1, failed at 65 MPa. Regarding the radial strain, the samples with thin walls (such as samples 8–10) recorded a sensitive response to the rate of pressure change compared to samples 1–7, which had a small rate of deformation, as can be seen in Figure 5. Also, there was a sudden increase in the hoop strain as the applied pressure got closer to the burst pressure, as was also noticed in a previous study [15]. Thus, the failure of the material propagates from the interior towards the exterior of the pipe once plastic deformation starts and quickly spreads to the outer surface, leading to the formation of cracks in the generator’s direction [1,15,16,17]. Figure 6 shows the variation curves of radial strain in the case of samples with β > 1.1 (Figure 6a), β = 1, and β < 1.1 (Figure 6b). The small pipes with ratios of more than 1.1 present a rigid–plastic behavior (Figure 6b), compared to specimens with ratios of less than 1.1, whose behaviour is elastic–plastic (Figure 6a). The specimen (8) found at the border between the two categories (β = 1.1) presents a rigid elastic–plastic behaviour.

Considering the way in which the small pipes failed, it can be observed that the failure appeared on the longitudinal axis, which is characteristic of robust material (Figure 7). The maximum failure pressure varies between 65 and 80 MPa for the thick-walled pipes and between 55 and 60 MPa for the thin-walled pipes. The micrographic analysis by optic microscopy highlights the propagation areas of the crack. It can be noticed that the ferrite (dark-coloured region) and pearlite (white-coloured region) structure is a typical arrangement of carbon steels (Figure 8). In Figure 8, it can be noticed that the fracture of the interior part differs from that in the exterior. Because the crack propagates from the inside, it is obvious that the material was subjected to circumferential tensile stress over the yielding limit. In accordance with [15], for a thin-walled pipe, once the wall thickness has completely undergone plastic yielding, the plastic strain increases rapidly, with a very small increase in pressure leading to a burst, as was experimentally obtained. When the elastic–plastic zone reaches the external surface of the pipe wall, the stability of the structure is lost, as can be seen in Figure 7 and Figure 8. So, it is recommended that under instantaneous overloading, the elastic–plastically deformed zone should not exceed the mean radius of the pipe wall, as argued in [18]. At the interface between the elastic and plastic zone in the wall of the pipe, the radial stress of the elastic region is equal to that of the plastic region [18,19]. In [19], the authors demonstrated numerically and experimentally that the inner wall of pipes yields first when the internal pressure increases to the yield strength of the material. 

### 3.2. Finite Element Analysis (FEA) of Strain and Stress States of Small Pipes

Two types of pipes were designed for finite element analysis (FEA): pipes with thick walls, characterized by β > 1.1, and thin walls, with β < 1.1. After importing the 3D model in Abaqus, the mechanical properties of steel E355 were used in the pre-processing step. From the point of view of boundary conditions, the “fixed at both ends” condition was applied and the model was loaded with an interior pressure of 80 N/mm^2^. Being a structure characterized by axial symmetry relative to the axis of revolution, and with the wall thickness reduced to a surface, the structure was discretised into triangular finite elements of the first order. The FEA results emphasized that subjecting a wall with β = 1.4 mm to internal pressure by 60 MPa led to tension in the pipe wall being increased by about 6 MPa (1.23%) comparison with pipes with β > 1.1. Being of the same material, OL52 (E355), the values of the maximum stresses are close in the two cases, both of which are up to the break stress (722 MPa) and below the yield stress (548 MPa) (Table 4). 

Regarding the displacement, it can be seen that in the longitudinal direction (x), the displacements (U_x_) are relatively low and approximately equal, regardless of the thickness of the pipe wall. In the radial direction (y = z), the values are equal with regard to the circular section of the pipe being influenced by the wall thickness. Such displacements are approximately two times higher in the circumferential direction for the pipe with wall thickness 2.2 times lower than the thick-walled pipe, as can be seen in Figure 9.

Comparing the results of the numerical model (FEM) and experiments, it can be noticed that their differences range between 0.3 and 0.6% (Figure 9). In Figure 10, the stress and displacement distribution are shown for each analysed case.

### 3.3. Cyclic Loading

Crack growth under variable loading has a crucial influence on structural life. The measured exterior diameters for each cycle were recorded in Table 5. First of all, it can be observed that the behaviour of the mini pipes differs depending on the ratio of the diameters, β. With the increase of the loading time from 30 to 60 s, the deformation increases in all three cases: in the case of sample A (with ratio β > 1.1), the increase is linear with an increasing number of cycles (Figure 11a); in the case of sample B (with ratio β = 1.1), the exterior diameter is increased step by step, which means that the energy is stored and dissipated in the wall of the pipe (Figure 11b); and in the case of sample C (β < 1.1), the increase is recorded as a combination of a linear and in-step trendline (Figure 11c). During the depressurisation, some residual radial strain is noticed, in which values depend on the period of loading and unloading; so, in the case of 30 s loading time, the value of deformation is smaller than the 60 s period of loading. 

The trend line of radial deformation with increasing cycle number and duration of exposure to internal pressure is presented in Figure 12. In all cases, the deformation does not exceed 0.02 mm for all exterior diameters (Figure 12a). With decreasing wall thickness of the mini pipes, the residual strain increased both in the case of the 30 s and 60 s loading cycles, but with different values, in accordance with the ratio between exterior and interior diameter, β, as can be seen in Figure 12a,b. The authors of [18] remarked that when the pipe is loaded elastic–plastically, a negligible increase of inner pressure caused a substantial increase of the elastic–plastically deformed zone. The variation of radial deformation at loading and unloading leads to increased plastic behaviour. In this situation, the hardening/softening phenomenon can be explained by change in the yield stress. Figure 13 shows that at the end of the test, samples with a ratio β higher than 1.1 record a radial strain deformation 100% smaller than that of samples with a ratio β equal to or smaller than 1.1 in the loading state. In the unloading state, sample C (with β < 1.1) has a residual strain four times higher than that of sample A. 

## 4. Conclusions

In this experimental study, the radial strains of mini honed pipes made from E355 steel were analysed in the elastic and plastic domain with both monotonic and cyclic loading. The monotonic tests were carried out to determine the fracture mode of pipe subjected to internal pressure without the influence of other types of loading. Due to the experimental setup, the internal pressurisation of the pipe led to radial strain with a relatively smaller axial strain (due to Poisson’s effect). The radial deformation depends on the thicknesses of the pipe wall, which in this study, were classified into three types in accordance with the ratio between the outer and inner diameters of the pipe. As mentioned in [15], for a thin-walled pipe (such as a pipe with the ratio β < 1.1) made from an ideal plastic material, once the wall thickness has completely undergone plastic yielding, the plastic strain increases rapidly with a very small increase in pressure, leading to a burst. The numerical simulation confirms the behaviour of pipes subjected to internal pressure. 

The cyclic tests with a maximum internal pressure around 25% lower than the burst pressure aim to assess the evolution of radial strain. The wall thickness and exterior diameter related to pipe radius play an important role in the high resistance of pipes. The variation of internal pressure in terms of cyclic loading leads to increasing the plastic deformation of pipe and risk of damage. Experiments show that the cyclic plastic characteristics of metallic materials are different from the monotonic ones. However, the behaviour of small pipes subjected to burst pressure depends on many factors, such as the type and intensity of loading, mechanical properties of pipe materials, thickness of wall, temperature, geometry of pipes, and environments. 

## Figures and Tables

**Figure 1 materials-12-02849-f001:**
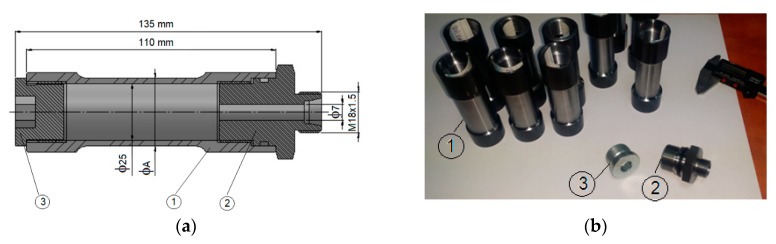
The shape of tested specimens: (**a**) The design of a sample section with a 2D view; (**b**) actual samples with joint bushing (1: steel honed pipe; 2: joint hex-nipple M118x1.5; 3: metallic cork with gasket).

**Figure 2 materials-12-02849-f002:**
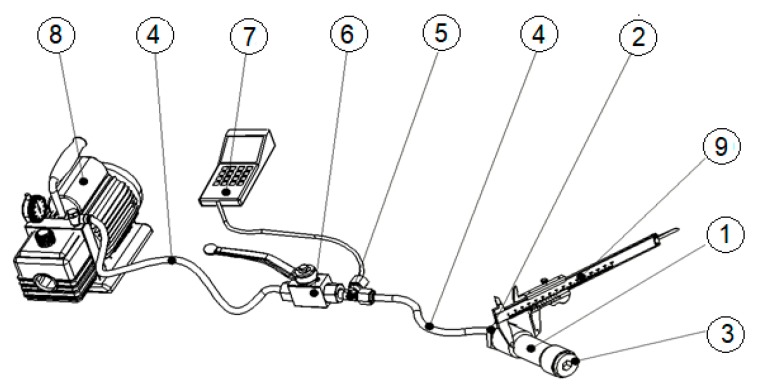
The experimental setup: 1: sample; 2: joint hex-nipple M118x1.5; 3: metallic cork with gasket; 4: hydraulic hose; 5: T-shaped connector; 6: faucet type KHB–D/4″; 7: supplementary manometer of 1000 bars; 8: electric pump; 9: electronic calliper.

**Figure 3 materials-12-02849-f003:**
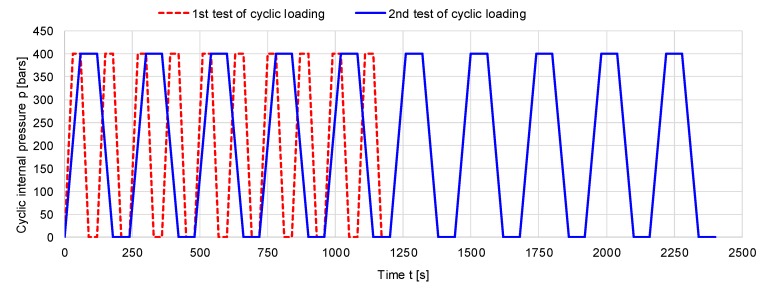
Variation of cyclic loading: zero-tension cycle.

**Figure 4 materials-12-02849-f004:**
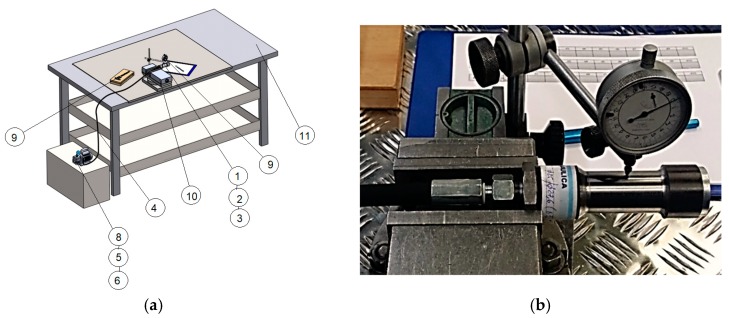
The experimental bench used for cyclic loading. (**a**) The design of the experimental setup (1: sample; 2: joint hex-nipple M118x1.5; 3: metallic cork with gasket; 4: hydraulic hose; 5: T-shaped connector; 6: faucet type KHB-D/4″; 8: electric pump; 9: measuring device; 10: sample fastener; 11: flat and rigid table; (**b**) measuring samples during the test.

**Figure 5 materials-12-02849-f005:**
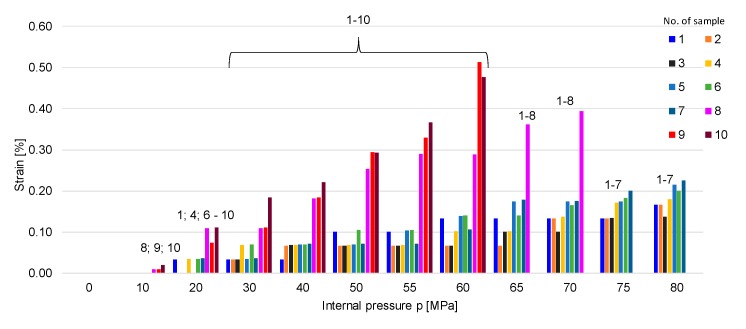
Increasing the exterior diameter of samples with applied internal pressure. The numbers written above the columns represent the numbers of samples which experienced radial strain when pressure was applied.

**Figure 6 materials-12-02849-f006:**
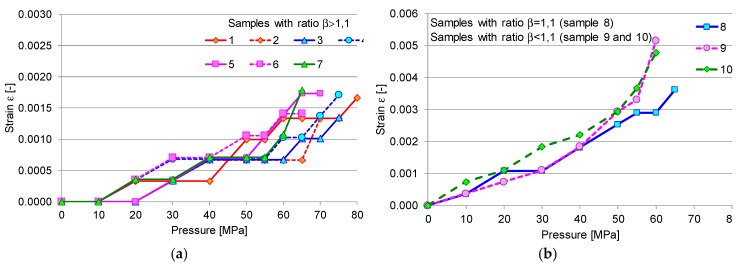
Variation of radial strain with increasing internal pressure: (**a**) for samples (1–7) with diameter ratios over 1.1; (**b**) for samples with an equal diameter ratio (sample 8) or ratios less than 1.1 (samples 9 and 10).

**Figure 7 materials-12-02849-f007:**
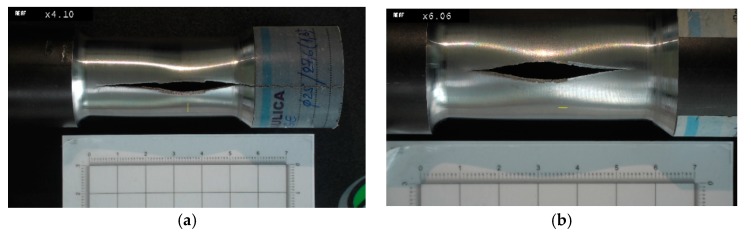
Rupture of pipe specimens due to internal pressure. (**a**) Sample 8; (**b**) Sample 9.

**Figure 8 materials-12-02849-f008:**
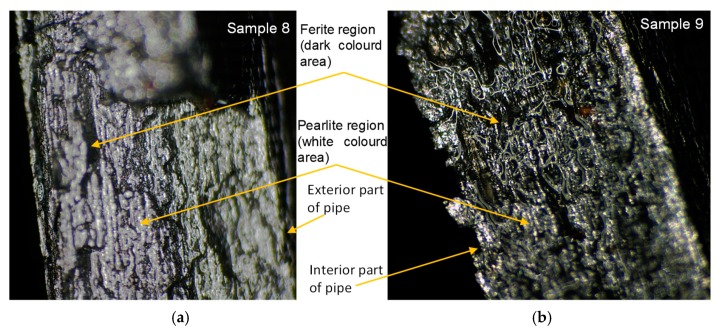
Optical micrograph images at 200× magnification of the failed area of the specimen: (**a**) sample 8; (**b**) sample 9.

**Figure 9 materials-12-02849-f009:**
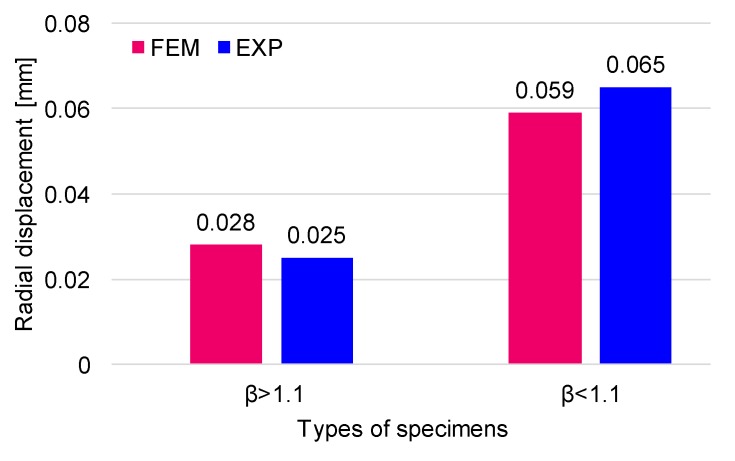
Comparison between finite element method (FEM) and experimental (denoted EXP) results.

**Figure 10 materials-12-02849-f010:**
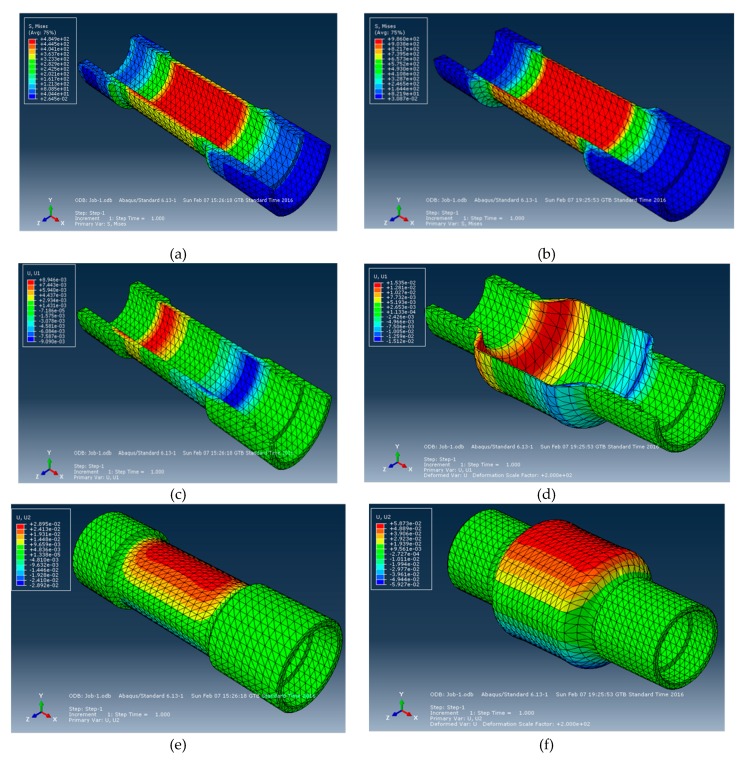
Stress and strain states of pipes obtained by finite element analysis (FEA): (**a**) von Mises stress in the case of a pipe with β > 1.1; (**b**) von Mises stress in the case of a pipe with β < 1.1; (**c**) displacement in the longitudinal direction (U_x_) in the case of a pipe with β > 1.1; (**d**) displacement in the longitudinal direction (U_x_) in the case of pipe with β < 1.1; (**e**) displacement in the y direction (U_y_) in the case of a pipe with β > 1.1; (**f**) displacement in the y direction (U_y_) in the case of a pipe with β < 1.1; (**g**) displacement in the z direction (U_z_) in the case of a pipe with β > 1.1; (**h**) displacement in the z direction (U_z_) in the case of a pipe with β < 1.1.

**Figure 11 materials-12-02849-f011:**
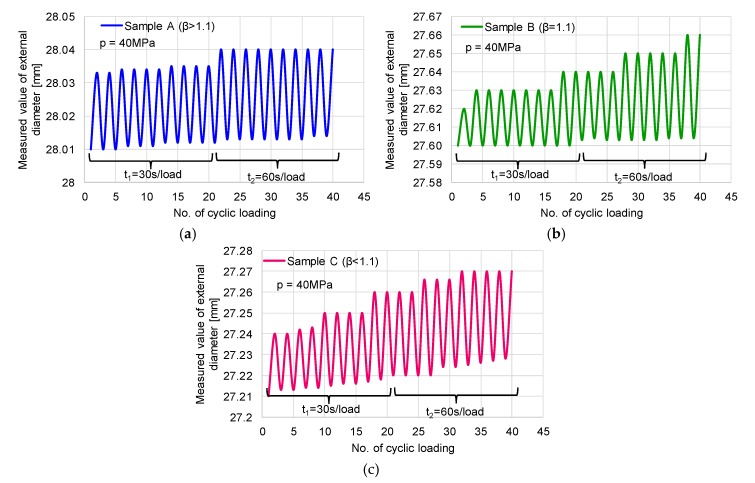
Cyclic variation of exterior diameters of tested samples: (**a**) Sample A; (**b**) sample B; (**c**) sample C.

**Figure 12 materials-12-02849-f012:**
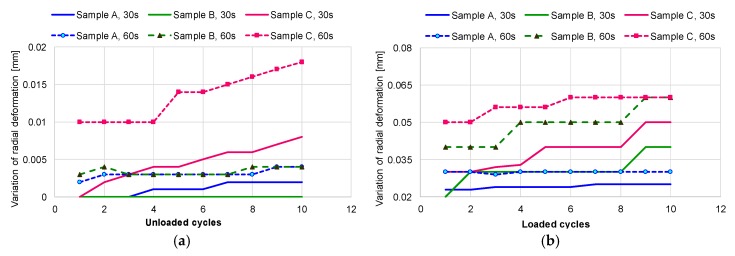
Variation of exterior diameter during test: (**a**) unloading cycles; (**b**) loading cycles.

**Figure 13 materials-12-02849-f013:**
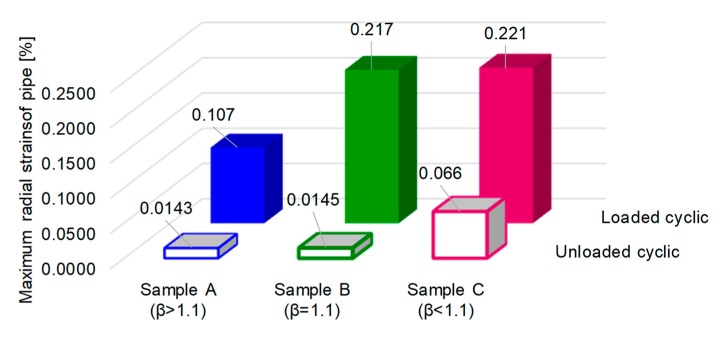
The final percentage of radial strain of tested pipes.

**Table 1 materials-12-02849-t001:** Geometrical characteristics of samples.

Sample Code	Interior Diameter Φ25 (mm)	Outside Diameter ΦA (mm)	Wall Thickness s_1_ = (ΦA − Φ25)/2 (mm)	Ratio B = ΦA/Φ25
1	25	30.01	2.505	1.20
2	25	29.99	2.495	1.19
3	25	29.60	2.300	1.18
4	25	29.20	2.100	1.16
5	25	28.80	1.900	1.15
6	25	28.40	1.700	1.13
7	25	28.00	1.500	1.12
8	25	27.60	1.300	1.10
9	25	27.22	1.110	1.08
10	25	27.21	1.105	1.08

**Table 2 materials-12-02849-t002:** Geometrical characteristics of samples tested to cyclic internal pressure.

Sample Code	Interior Diameter Φ25 (mm)	Outside Diameter ΦA (mm)	Wall Thickness s_1_ = (ΦA − Φ25)/2 (mm)	Ratio β = ΦA/Φ25
A	25	28.010	1.505	1.12
B	25	27.600	1.300	1.10
C	25	27.210	1.105	1.08

**Table 3 materials-12-02849-t003:** The measured values of exterior diameter.

Samples	Pressure (MPa)											
	0	10	20	30	40	50	55	60	65	70	75	80
1	30.01	30.01	30.02	30.02	30.02	30.04	30.04	30.05	30.05	30.05	30.05	30.06
2	29.99	29.99	29.99	30.00	30.01	30.01	30.01	30.01	30.01	30.03	30.03	30.04
3	29.60	29.60	29.60	29.61	29.62	29.62	29.62	29.62	29.63	29.63	29.64	29.64
4	29.20	29.20	29.21	29.22	29.22	29.22	29.22	29.23	29.23	29.24	29.25	29.25
5	28.80	28.80	28.80	28.81	28.82	28.82	28.83	28.84	28.85	28.85	28.85	28.86
6	28.40	28.40	28.41	28.42	28.42	28.43	28.43	28.44	28.44	28.45	28.45	28.46
7	28.00	28.00	28.01	28.01	28.02	28.02	28.02	28.03	28.05	28.05	28.06	28.06
8	27.60	27.61	27.63	27.63	27.65	27.67	27.68	27.68	27.70	27.71	burst	burst
9	27.22	27.23	27.24	27.25	27.27	27.30	27.31	27.36	burst	burst	burst	burst
10	27.21	27.23	27.24	27.26	27.27	27.29	27.31	27.34	burst	burst	burst	burst

**Table 4 materials-12-02849-t004:** The results of finite element analysis (FEA) simulation.

Type of Pipe	Ratio β	Pressure P (N/mm^2^)	Maximum Stress: Von Mises σ_max_ (MPa)	Displacement U_tot_ (mm)	Displacement		
					U_x_ (mm)	Uy (mm)	Uz (mm)
Thick wall	β > 1.1	80	485	0.0292	0.009	0.028	0.028
Thin wall	β < 1.1	65	491	0.0601	0.015	0.059	0.059

**Table 5 materials-12-02849-t005:** The results of the cyclic loading test for all three types of specimens.

**t_1_ = 30 s**	**Diameter Measured after Each Cycle ΦA (mm)** **Sample A** (β > 1.1)									
	**d1**	**d2**	**d3**	**d4**	**d5**	**d6**	**d7**	**d8**	**d9**	**d10**
p_active_ *	28.033	28.033	28.034	28.034	28.034	28.034	28.035	28.035	28.035	28.035
p_inactive_ *	28.010	28.010	28.010	28.011	28.011	28.011	28.012	28.012	28.012	28.012
**t_2_ = 60 s**	**d11**	**d12**	**d13**	**d14**	**d15**	**d16**	**d17**	**d18**	**d19**	**d20**
p_active_ *	28.040	28.040	28.040	28.040	28.040	28.040	28.040	28.040	28.040	28.040
p_inactive_ *	28.012	28.013	28.013	28.013	28.013	28.013	28.013	28.013	28.014	28.014
**t_1_ = 30 s**	**Sample B** (β = 1.1)									
p_active_ *	27.620	27.630	27.630	27.630	27.630	27.630	27.630	27.630	27.640	27.640
p_inactive_ *	27.600	27.600	27.600	27.600	27.600	27.600	27.600	27.600	27.600	27.600
**t_2_ = 60 s**	**d11**	**d12**	**d13**	**d14**	**d15**	**d16**	**d17**	**d18**	**d19**	**d20**
p_active_ *	27.640	27.640	27.640	27.650	27.650	27.650	27.650	27.650	27.660	27.660
p_inactive_ *	27.603	27.604	27.603	27.603	27.603	27.603	27.603	27.604	27.604	27.604
**t_1_ = 30 s**	**Sample C** (β < 1.1)									
p_active_ *	27.240	27.240	27.242	27.243	27.250	27.250	27.250	27.250	27.260	27.260
p_inactive_ *	27.21	27.212	27.213	27.214	27.214	27.215	27.216	27.216	27.217	27.218
**t_2_ = 60 s**	**d11**	**d12**	**d13**	**d14**	**d15**	**d16**	**d17**	**d18**	**d19**	**d20**
p_active_ *	27.260	27.260	27.266	27.266	27.266	27.270	27.270	27.270	27.270	27.270
27.220	27.220	27.220	27.220	27.224	27.224	27.225	27.226	27.227	27.228	27.220

* p_active_ denotes during pressure loading, and p_inactive_ denotes during unloading.

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
