# Peer review of "The Effect of Internal Pressure on Radial Strain of Steel Pipe Subjected to Monotonic and Cyclic Loading"

_materials, 2019, doi:10.3390/ma12182849_

Round 1

Reviewer 1 Report

This paper put an effort to show the influence of internal pressure on the
elastic and plastic behavior of E355 steel pipes.

Even though the paper is well presented with too many useful experimental results. The paper have some grammatical mistakes which needs to modified.

Lines 60-64: rewrite the sentences. little confusing. There is no proper connections.

Line 90: Is it 100 to 100 bars (or) 100 to 1000 bars. Make sure that the range is right.

Line 98: Check the unit. Properly place a superscript in N/mm2

Table 2. remove extra equal = symbol from ratio column.

Line 115: Spelling mistake "Regarding"

Table 3: remove p after Pressure. It is not necessary.

Line 134: Spelling mistake "Yield"

Line 166: Correct grammatical mistake. "In Figure 8 are presented"

Please go through the paper and understand the grammatical errors and then do the modifications.

Thank you.

Author Response

First we would like to thank the reviewers for carefully going through the manuscript and providing helpful suggestions for its improvement. Thanks to their constructive comments, we are able to present clearly and better version than the original manuscript. All the comments of the reviewers have been considered. In particular, the following changes have been made according to the reviewers' suggestions, highlighted by red colour in the manuscript.

Reviewer 2 Report

Dear Authors,

The experimental tests of mini honed pipes subjected to the internal pressure, with different thicknesses, made of E355 steel are described in the paper. The analyses are based on the measurements of the external diameter of the pipe subjected to monotonic and cyclic loadings. The paper is interesting, however, in my opinion at the moment is rather short communication from the experimental tests. Because of this, it requires comparison of the obtained results with theoretical models.

Major and minor remarks:

p.1. A few analytical methods are mentioned in the “1. Introduction” and in “4. Conclusions” it was summarized that “behavior of small pipes subjected to burst pressure in the absence of externally applied axial stress is in good agreement with prediction models developed in literature.”, however such analytical analyses are not presented and not discussed in the paper. Such theoretical analysis should be supplemented in the paper for both monotonic and cyclic loadings.

p.2. Lines 103-104 “In this test, three types of samples were investigated in accordance with ratio between outer and inner diameters, denoted β: sample A with β>1.1; sample B with β≈1.1 and sample C with β<1.1.” Exact values should be given instead approximate values.

p.3. Table 2. Description “B==” of the ratio should be corrected.

p.4. lines 112-114 “It can be noticed that the samples 8, 9 and 10 with ratio between exterior diameter and interior diameter less than 1.5, failed befor to 65 MPa (in case of samples 9 and 10) and 75 MPa – in case of sample 8.”. This statement is not clear. All investigated samples (not only 8,9 and 10), given in Table 1, have ratio between exterior diameter and interior diameter less than 1.5. Moreover the ratio “beta” β should be given in the Table 1.

p.5. Line 114 “befor” should be corrected

p.6. Figure 4: I propose to add information that numbers above the diagrams as well as on the left hand-side is the sample number.

p.7. Lines 138-139: “So, it is recommended that under instantaneous overloading the elasto plastically deformed zone should not exceed the mean radius of the pipe wall as [18] argues.” More information about this conclusion will be interesting.

p.8. Table 4. I propose to add information about “beta” in the table (i.e. after Sample X, β= X)for all three samples A, B and C.

p.9. Lines 153-157 ” The behavior of sample B which recorded a decreases of diameter during unloading time, can be explained by a sorption phenomenon occurred during depressurization due to the presence of an air bubble which has led to the phenomenon vacuum pressure-vacuum [4, 6]. All three samples (the partition walls, the thick and thin) were subjected to stress under similar conditions, so how to determine the curves of variation of the strains was performed in the same manner.” It is difficult to understand. Please clarify. The obtained results (Fig.7) as well as the assumption (given above by the Authors) seem, that all three samples were not subjected to stress under similar conditions. In such situation the sample B should be studied more carefully (by i.e. finite element analysis) in order to investigate the problem of the decreasing of the external diameter under internal pressure. Such explanation is not sufficient.

Kind Regards

Author Response

(The authors gave the same response as above.)

Round 2

Reviewer 2 Report

Dear Authors,

Thank you for your revision. I have one remark:

Numbering of figures (and references to the figures in the text) should be corrected and carefully checked. There are figures 10 and 12, but Figure 11 is not used.

Kind Regards.

Author Response

Thank you for remark.

We corrected the figure's number and references to them in text.

Best regards.
